# Byzantine Stochastic Gradient Descent

**Dan Alistarh**[*]
IST Austria
dan.alistarh@ist.ac.at

**Zeyuan Allen-Zhu**[*]
Microsoft Research AI
zeyuan@csail.mit.edu

**Jerry Li**[*]
Simons Institute
jerryzli@berkeley.edu

## Abstract

This paper studies the problem of distributed stochastic optimization in an adversarial setting where, out of $m$ machines which allegedly compute stochastic gradients every iteration, an $\alpha$-fraction are Byzantine, and may behave adversarially. Our main result is a variant of stochastic gradient descent (SGD) which finds $\varepsilon$-approximate minimizers of convex functions in $T = \widetilde{O}\big(\frac{1}{\varepsilon^2 m} + \frac{\alpha^2}{\varepsilon^2}\big)$ iterations. In contrast, traditional mini-batch SGD needs $T = O\big(\frac{1}{\varepsilon^2 m}\big)$ iterations, but cannot tolerate Byzantine failures. Further, we provide a lower bound showing that, up to logarithmic factors, our algorithm is information-theoretically optimal both in terms of sample complexity and time complexity.

## 1 Introduction

Machine learning applications are becoming increasingly decentralized, either because data is naturally distributed—in applications such as *federated learning* [17]—or because data is partitioned across machines to parallelize computation, e.g. [2]. *Fault-tolerance* is a critical concern in such distributed settings. Machines in a data center may crash, or fail in unpredictable ways; even worse, in some settings one must be able to tolerate a fraction of *adversarial/faulty* workers, sending corrupted or even malicious data. This *Byzantine* failure model—where a small fraction of bad workers are allowed to behave arbitrarily—has a rich history in distributed computing [19]. By contrast, the design of machine learning algorithms which are robust to such Byzantine failures is a relatively recent topic, but is rapidly becoming a major research direction at the intersection of machine learning, distributed computing, and security.

We measure algorithms in this setting against two fundamental criteria: *sample complexity*, which requires high accuracy from few data samples, and *computational complexity*, i.e. preserving the runtime speedups achieved by distributing computation. These criteria should hold even under adversarial conditions. Another important consideration in the design of these algorithms is that they should remain useful in high dimensions.

**System Model.** We study stochastic optimization in the Byzantine setting. We assume an unknown distribution $\mathcal{D}$ over functions $\mathbb{R}^d \rightarrow \mathbb{R}$, and wish to minimize $f(x) := \mathbb{E}_{s \sim \mathcal{D}}[f_s(x)]$.

We consider a standard setting with $m$ workers and a master (coordinator), where an $\alpha$-fraction of the workers may be Byzantine, with $\alpha < 1/2$. Each worker has access to $T$ sample functions from the distribution $\mathcal{D}$. We proceed in iterations, structured as follows: workers first perform some local computation, then synchronously send information to the master, which compiles the information and sends new information to the workers. At the end, the master should output an approximate minimizer of the function $f$.

While our negative results will apply for this general setting, our algorithms will be expressed in the standard framework of distributed stochastic gradient methods: in each iteration $k$, the master broadcasts the current iterate $x_k \in \mathbb{R}^d$ to worker machines, and each worker is supposed to compute a stochastic gradient at $x_k$ and return it to the master. A *good* worker returns returns $\nabla f_s(x_k)$ for a

---

[*]Authors in alphabetical order. Full version can be found on https://arxiv.org/abs/1803.08917.

random sample $s \sim \mathcal{D}$, but a Byzantine worker machine may adversarially return any vector. This stochastic optimization framework is general and very well studied, and captures many important problems such as regression, learning SVMs, logistic regression, and training deep neural networks. Traditional methods such as mini-batch *stochastic gradient descent (SGD)* are vulnerable to even a single Byzantine failure. Our results are presented in the master-worker distribution model, but can be generalized to a coordinator-free distributed setting using standard techniques [12], assuming authenticated point-to-point channels.

In this setting, *sample complexity* is measured as the number of functions $f_s(\cdot)$ we accessed. Since every machine gets one sample per iteration, minimizing sample complexity is equivalent to minimizing the number of iterations. *Time complexity* is determined by the number of iterations.

**Our Results.** In this work, we study the convex formulation of this Byzantine stochastic optimization problem: we assume $f(x)$ is convex, although each of the functions $f_s(x)$ may not necessarily be convex. We provide the first algorithms that, in the presence of Byzantine machines, guarantee the following, up to logarithmic and lower-order terms:

(1) achieve optimal sample complexity,

(2) achieve optimal number of stochastic gradient computations,

(3) match the sample and time complexities of traditional SGD as $\alpha \to 0$, and

(4) achieve (1)-(3) even as the dimension grows, without losing additional dimension factors.

In addition, our algorithms are optimally-robust, supporting a fraction of $\alpha < 1/2$ Byzantine workers. Despite significant recent interest, e.g. [6, 8, 13, 26, 27, 30, 31], to the best of our knowledge, prior to our work there were no algorithms for stochastic optimization in high dimensions that achieved any of the four objectives highlighted above. Previous algorithms either provided weak robustness guarantees, or had sample or time complexities which degrade polynomially with the dimension $d$ or with the error $\varepsilon$.

**Technical Contribution.** A direct way to deal with Byzantine workers is to perform a robust aggregation step to compute gradients, such as median of means: for each (good) worker machine $i \in [m]$, whenever a query point $x_k$ is provided by the master, the worker takes $n$ stochastic gradient samples and computes their average, which we call $v_i$. If $n = \widetilde{\Theta}(\varepsilon^{-2})$, one can show that for each good machine $i$, it holds that $\|v_i - \nabla f(x_k)\| \le \varepsilon$ with high probability. Therefore, in each iteration $k$, we can determine a vector $v_{\mathsf{med}} \in \{v_1, \ldots, v_m\}$ satisfying $\|v_{\mathsf{med}} - \nabla f(x_k)\| \le 2\varepsilon$, and move in the negative direction of $v_{\mathsf{med}}$.

However, the above idea requires too many computations of stochastic gradients. In the non-strongly convex setting, each worker machine needs to compute $\varepsilon^{-2}$ stochastic gradients per iteration, and the overall number of iterations will be at least $\varepsilon^{-1}$. This is because, even when $f_s(x) = f(x)$ and $\alpha = 0$, gradient descent converges in $\varepsilon^{-1}$ iterations. This amounts to a sample complexity of linear dependency in $\varepsilon^{-3}$.

We take a different approach. We run the algorithm for $T$ iterations, where each machine $i \in [m]$ only computes *one* stochastic gradient per iteration. Let $v_i^{(k)}$ be the stochastic gradient allegedly computed by machine $i \in [m]$ at iteration $k \in [T]$. By martingale concentration, $B_i := (v_i^1 + \cdots + v_i^{(T)})/T$ should concentrate around $B_\star := (\nabla f(x_1) + \cdots + \nabla f(x_T))/T$ for each good machine $i$, up to an additive error $\frac{1}{\sqrt{T}}$. Hence, if $\|B_i - B_\star\| > 1/\sqrt{T}$ for machine $i$, we can safely declare that $i$ is Byzantine.

Two non-trivial technical obstacles remain. First, we cannot restart the algorithm every time we discover a new Byzantine machine, since that would ruin its time complexity. Second, Byzantine machines may successfully "disguise" themselves by not violating the above criterion.

To address the first issue, we keep track of the quantity

$$B_i^{(k)} := \frac{v_i^1 + \cdots + v_i^{(k)}}{k}$$

at each step $k$; if a machine strays away too much from $B_\star^{(k)}$, it is labeled as Byzantine, and removed from future consideration. We prove that restarts are not necessary. For the second problem, we

| algorithm | # sampled functions per machine | total work | per-iteration per-machine work |
|---|---|---|---|
| SGD ($\alpha = 0$ only)  (folklore) | $O\big(\frac{1}{\varepsilon} + \frac{1}{\varepsilon^2 m}\big)$ | $O\big(\frac{m}{\varepsilon} + \frac{1}{\varepsilon^2}\big)$ | 1 |
| ByzantineSGD  (Theorem 3.2) | $\widetilde{O}\big(\frac{1}{\varepsilon} + \frac{1}{\varepsilon^2 m} + \frac{\alpha^2}{\varepsilon^2}\big)$ | $\widetilde{O}\big(\frac{m}{\varepsilon} + \frac{1}{\varepsilon^2} + \frac{\alpha^2 m}{\varepsilon^2}\big)$ | 1 |
| GD ($\alpha = 0$ only)  (folklore) | $\widetilde{O}\big(1 + \frac{1}{\varepsilon^2 m}\big)$ | $\widetilde{O}\big(\frac{m}{\varepsilon} + \frac{1}{\varepsilon^3}\big)$ | $1 + \widetilde{O}\big(\frac{1}{\varepsilon^2 m}\big)$ |
| Median-GD  (Yin et al. [31]) | $\widetilde{O}\big(1 + \frac{d}{\varepsilon^2 m} + \frac{\alpha^2}{\varepsilon^2}\big)$ | $\widetilde{O}\big(\frac{m}{\varepsilon} + \frac{d}{\varepsilon^3} + \frac{\alpha^2 m}{\varepsilon^3}\big)$ | $1 + \widetilde{O}\big(\frac{d}{\varepsilon^2 m} + \frac{\alpha^2}{\varepsilon^2}\big)$ |
| folklore  (c.f. [29, Theorem 11]) | $\Omega\big(\frac{1}{\varepsilon^2 m}\big)$ | $\Omega\big(\frac{1}{\varepsilon^2}\big)$ | |
| this paper  (Theorem 4.3) | $\Omega\big(\frac{1}{\varepsilon^2 m} + \frac{\alpha^2}{\varepsilon^2}\big)$ | $\Omega\big(\frac{1}{\varepsilon^2} + \frac{\alpha^2 m}{\varepsilon^2}\big)$ | |
| $\Uparrow$ convex $\Uparrow$ | | $\Downarrow$ $\sigma$-strongly convex $\Downarrow$ | |
| SGD ($\alpha = 0$ only)  (folklore) | $O\big(\frac{1}{\sigma} + \frac{1}{\sigma \varepsilon m}\big)$ | $O\big(\frac{m}{\sigma} + \frac{1}{\sigma \varepsilon}\big)$ | 1 |
| ByzantineSGD  (Theorem 3.4) | $\widetilde{O}\big(\frac{1}{\sigma} + \frac{1}{\sigma \varepsilon m} + \frac{\alpha^2}{\sigma \varepsilon}\big)$ | $\widetilde{O}\big(\frac{m}{\sigma} + \frac{1}{\sigma \varepsilon} + \frac{\alpha^2 m}{\sigma \varepsilon}\big)$ | 1 |
| GD ($\alpha = 0$ only)  (folklore) | $O\big(1 + \frac{1}{\sigma \varepsilon m}\big)$ | $O\big(\frac{m}{\sigma} + \frac{1}{\sigma^2 \varepsilon}\big)$ | $1 + O\big(\frac{1}{\sigma \varepsilon m}\big)$ |
| Median-GD  (Yin et al. [31]) | $\widetilde{O}\big(1 + \frac{d}{\sigma \varepsilon m} + \frac{\alpha^2}{\sigma \varepsilon}\big)$ | $\widetilde{O}\big(\frac{m}{\sigma} + \frac{d}{\sigma^2 \varepsilon} + \frac{\alpha^2 m}{\sigma^2 \varepsilon}\big)$ | $1 + \widetilde{O}\big(\frac{d}{\sigma \varepsilon m} + \frac{\alpha^2}{\sigma \varepsilon}\big)$ |
| folklore  (c.f. [29, Appendix C.5]) | $\Omega\big(\frac{1}{\sigma \varepsilon m}\big)$ | $\Omega\big(\frac{1}{\sigma \varepsilon}\big)$ | |
| this paper  (Theorem 4.4) | $\Omega\big(\frac{1}{\sigma \varepsilon m} + \frac{\alpha^2}{\sigma \varepsilon}\big)$ | $\Omega\big(\frac{1}{\sigma \varepsilon} + \frac{\alpha^2 m}{\sigma \varepsilon}\big)$ | |

Table 1: Comparison of Byzantine optimization for smooth convex minimization $f(x) = \mathbb{E}_{s \sim \mathcal{D}}[f_s(x)]$.
    **Remark 1.** In this table, we have hidden parameters $L$ (smoothness), $\mathcal{V}$ (variance), and $D$ (diameter). The goal is to achieve $f(x) - f(x^*) \le \varepsilon$, and $\sigma$ is the strong convexity parameter of $f(x)$.
    **Remark 2.** "# sampled functions" is the number of $f_s(\cdot)$ to sample for each machine.
    **Remark 3.** "total/per-iteration work" is in terms of the # of stochastic gradient computations $\nabla f_s(\cdot)$.

construct a similar "safety" criterion, in terms of the sequence

$$A_i^{(k)} := \frac{\langle v_i^{(1)}, x_1 - x_0 \rangle + \cdots + \langle v_i^{(k)}, x_k - x_0 \rangle}{k}.$$

We prove that good machines will satisfy both criteria; more importantly, any *Byzantine* machine which satisfies both of them must have negligible negative influence in the algorithm's convergence.

**Related Work.** The closest work to ours is the concurrent and independent work of Yin et al. [31]. They consider a similar Byzantine model, but for *gradient descent (GD)*. In their algorithm, *each* of the $m$ machines receives $n$ samples of functions upfront. In an iteration $k$, machine $i$ allegedly computes $n$ stochastic gradients at point $x_k$ and averages them (the $n$ stochastic gradients are taken with respect to the $n$ sampled functions stored on machine $i$). Then, their proposed algorithm aggregates all the average vectors from the $m$ machines, and performs a coordinate-wise median operation to determine the descent direction. In contrast, our algorithm is a *Byzantine variant of SGD*: a total of $Tm$ functions are sampled and a total of $Tm$ stochastic gradient computations are performed. To be robust against Byzantine machines, they average stochastic gradients within a single iteration and compare them across machines. In contrast, we average stochastic gradients (and other quantities) *across iterations*.

Further, in terms of sample complexity (i.e., the number of functions $f_s(\cdot)$ to be sampled), their algorithm's complexity is higher by a linear factor in the dimension $d$ (see Table 1). This is in large part due to their coordinate-wise median operation. In high dimensions, this leads to sub-optimal statistical rates. In terms of total computational complexity, each iteration of Yin et al. [31] requires a full pass over the (sampled) dataset. In contrast, an entire run of ByzantineSGD requires only one pass. Finally, their algorithm works under a weaker set of assumptions than ours. They assumed that the stochastic error in gradients (namely, $\nabla f_s(x) - \nabla f(x)$) has bounded variance and skewness; in contrast, we only assume that $\nabla f_s(x) - \nabla f(x)$ is bounded with probability 1. Our stronger assumption (which is standard) turns out to simplify our algorithm and analysis. We leave it as future work to extend ByzantineSGD to bounded skewness.

Yin et al. [31] also provided a lower bound in terms of sampling complexity — the number of functions $f_s(\cdot)$ needed to be sampled in the presence of Byzantine machines. When translated to

our language, the result is essentially the same as the strongly convex part of Theorem 4.4. The results in this paper are the first to cover the case of non-strongly convex functions.

**Byzantine Stochastic Optimization.** There has been a lot of recent work on Byzantine stochastic optimization, and in particular, SGD [6, 8, 13, 26, 27, 30]. One of the first references to consider this setting is Feng et al. [13], which investigated distributed PCA and regression in the Byzantine distributed model. Their general framework has each machine running a robust learning algorithm locally, and aggregating results via a robust estimator. However, the algorithm requires careful parametrization of the sample size at each machine to obtain good error bounds, which renders it suboptimal with respect to sample complexity. Our work introduces new techniques which address both these limitations. Su and Vaidya [26, 27] consider a similar setting: in Su and Vaidya [26] they focus on the single-dimensional ($d = 1$) case, whereas Su and Vaidya [27] considers the multi-dimensional setting, but only consider a restricted family of consensus-based algorithms.

Blanchard et al. [6] propose a general Byzantine-resilient gradient aggregation rule called *Krum* for selecting a valid gradient update. This rule has local complexity $O(m^2(d + \log m))$, which makes it relatively expensive to compute when the $d$ and/or $m$ are large. Moreover, in each iteration the algorithm chooses a gradient corresponding to a constant number of correct workers, so the scheme does not achieve speedup with respect to the number of distributed workers, which negates an important benefit of distributed training. Xie et al. [30] consider gradient aggregation rules in a *generalized* Byzantine setting where a subset of the messages sent between servers can be corrupted. The complexity of their selection rule can be as low as $\widetilde{O}(dm)$, but their approach is far from sample-optimal. Chen et al. [8] leverage the geometric median of means idea in a novel way, which allows it to be significantly more sample-efficient, and applicable for a wider range of parameters. At the same time, their technique only applies in the strongly convex setting, and is suboptimal in terms of convergence rate by a factor of $\sqrt{\alpha m}$.

**Adversarial Noise.** Optimization and learning in the presence of adversarial noise is a well-studied problem [4, 5, 15, 20, 22, 28]. Recently, efficient algorithms for high dimensional optimization which are tolerant to a small fraction of adversarial corruptions have been developed [1, 7, 11, 16, 24], building on new algorithms for high dimensional robust statistics [5, 7, 9, 18]. This setting is different from ours. For instance, in their setting, there are statistical barriers so that no algorithm can achieve an optimization error below some fixed threshold, no matter how many samples are taken. In contrast, in the current Byzantine setting, the adversarial corruptions can only occur in a fraction of the machines (as opposed to each machine having some adversarial corruptions). For this reason, our results do not extend to their scenario.

## 2 Preliminaries

Throughout this paper, we denote by $\|\cdot\|$ the Euclidean norm and $[n] := \{1, 2, \ldots, n\}$. We reiterate some definitions regarding strong convexity, smoothness, and Lipschitz continuity (for other equivalent definitions, see Nesterov [21]).

**Definition 2.1.** *For a differentiable function $f : \mathbb{R}^d \to \mathbb{R}$,*

- *$f$ is $\sigma$-strongly convex if $\forall x, y \in \mathbb{R}^d$, it satisfies $f(y) \geq f(x) + \langle \nabla f(x), y - x \rangle + \frac{\sigma}{2} \|x - y\|^2$.*

- *$f$ is $L$-Lipschitz smooth (or $L$-smooth for short) if $\forall x, y \in \mathbb{R}^d$, $\|\nabla f(x) - \nabla f(y)\| \leq L\|x - y\|$.*

- *$f$ is $G$-Lipschitz continuous if $\forall x \in \mathbb{R}^d$, $\|\nabla f(x)\| \leq G$.*

**Byzantine Convex Stochastic Optimization.** We let $m$ be number of worker machines and assume at most an $\alpha$ fraction of them are Byzantine for $\alpha \in \left[0, \frac{1}{2}\right)$. We denote by good $\subseteq [m]$ the set of good (i.e. non-Byzantine) machines. Obviously, the algorithm does not know good.

We let $\mathcal{D}$ be a distribution over (not necessarily convex) functions $f_s : \mathbb{R}^d \to \mathbb{R}$. Our goal is to approximately minimize the following objective:

$$\min_{x \in \mathbb{R}^d} \left\{ f(x) := \mathbb{E}_{s \sim \mathcal{D}}[f_s(x)] \right\}, \tag{2.1}$$

where we assume $f$ is convex. In each iteration $k = 1, 2, \ldots, T$, the algorithm is allowed to specify a point $x_k$ and query $m$ machines. Each machine $i \in [m]$ gives back a vector $\nabla_{k,i} \in \mathbb{R}^d$ satisfying

**Assumption 2.2.** *For each iteration $k \in [T]$ and for every $i \in$ good, we have $\nabla_{k,i} = \nabla f_s(x_k)$ for a random sample $s \sim \mathcal{D}$, and $\|\nabla_{k,i} - \nabla f(x_k)\| \leq \mathcal{V}$.*

*Remark* 2.3. For each $k \in [T]$ and $i \notin$ good, the vector $\nabla_{k,i}$ can be adversarially chosen and may depend on $\{\nabla_{k',i}\}_{k' \leq k, i \in [m]}$. In particular, the Byzantine machines can even collude in an iteration.

The next fact is completely classical (for projected mirror descent).

**Fact 2.4.** *If $x_{k+1} = \arg\min_{y:\; \|y-x_1\| \leq D}\{\frac{1}{2}\|y - x_k\|^2 + \eta\langle\xi, y - x_k\rangle\}$, then $\forall u\colon \|u - x_1\| \leq D$:*

$$\langle\xi, x_k - u\rangle \leq \langle\xi, x_k - x_{k+1}\rangle - \frac{\|x_k - x_{k+1}\|^2}{2\eta} + \frac{\|x_k - u\|^2}{2\eta} - \frac{\|x_{k+1} - u\|^2}{2\eta} \; .$$

# 3 Description and Analysis of ByzantineSGD

Without loss of generality, in this section we assume that we are given a starting point $x_1 \in \mathbb{R}^d$ and want to solve the following more general problem:[2]

$$\min_{\|x-x_1\| \leq D}\left\{f(x) := \mathbb{E}_{s \sim \mathcal{D}}[f_s(x)]\right\} \; . \tag{3.1}$$

We denote by $x^*$ an arbitrary minimizer to Problem (3.1).

Our algorithm `ByzantineSGD` is formally stated in Algorithm 1. In each iteration at point $x_k$, `ByzantineSGD` tries to identify a set $\mathsf{good}_k$ of "candidate good" machines, and then perform stochastic gradient update only with respect to $\mathsf{good}_k \subseteq [m]$, by using direction $\xi_k := \frac{1}{m}\sum_{i \in \mathsf{good}_k}\nabla_{k,i}$.

The way $\mathsf{good}_k$ is maintained is by constructing two "estimation sequences". Namely, for each machine $i \in [m]$, we maintain a real value $A_i = \sum_{t=1}^{k}\langle\nabla_{t,i}, x_t - x_1\rangle$ and a vector $B_i = \sum_{t=1}^{k}\nabla_{t,i}$. Then, we denote by $A_{\mathsf{med}}$ the median of $\{A_1, \ldots, A_m\}$ and $B_{\mathsf{med}}$ some "vector median" of $\{B_1, \ldots, B_m\}$. We also define $\nabla_{\mathsf{med}}$ to be some "vector median" of $\{\nabla_{k,1}, \ldots, \nabla_{k,m}\}$. For instance for $\{\nabla_{k,1}, \ldots, \nabla_{k,m}\}$, our vector median is defined as follows. We select $\nabla_{\mathsf{med}}$ to be any $\nabla_{k,i}$ as long as $\big|\{j \in [m]\colon \|\nabla_{k,j} - \nabla_{k,i}\| \leq 2\mathcal{V}\}\big| > m/2$. Such an index $i \in [m]$ can be efficiently computed because our later lemmas shall ensure that at least $(1 - \alpha)m$ indices in $[m]$ are valid choices for $i$. Therefore, one can for instance guess a random index $i$ and verify whether it is valid. In expectation at most 2 guesses are needed, so finding these quantities can be done in linear time.

Starting from $\mathsf{good}_0 = [m]$, we define $\mathsf{good}_k$ to be all the machines $i$ from $\mathsf{good}_{k-1}$ whose $A_i$ is $\mathfrak{T}_A$-close to $A_{\mathsf{med}}$, $B_i$ is $\mathfrak{T}_B$-close to $B_{\mathsf{med}}$, and $\nabla_{k,i}$ is $4\mathcal{V}$-close to $\nabla_{\mathsf{med}}$. We will prove that if the thresholds $\mathfrak{T}_A$ and $\mathfrak{T}_B$ are chosen appropriately, then $\mathsf{good}_k$ always contains all machines in good.

**Bounding the Error.** As we shall see, the "error" incurred by `ByzantineSGD` contains two parts:

$$\mathsf{Error}_1 := \sum_{k \in [T]}\sum_{i \in \mathsf{good}_k}\langle\nabla_{k,i} - \nabla f(x_k), x_k - x^*\rangle$$

and

$$\mathsf{Error}_2 := \frac{1}{T}\sum_{k \in [T]}\left\|\frac{1}{m}\sum_{i \in \mathsf{good}_k}\left(\nabla_{k,i} - \nabla f(x_k)\right)\right\|^2.$$

$\mathsf{Error}_1$ is due to the bias created by the stochastic gradient (of good machines) and the adversarial noise (of Byzantine machines); while $\mathsf{Error}_2$ is the variance of using $\xi_k$ to approximate $\nabla f(x_k)$.

As we shall see, $\mathsf{Error}_2$ is almost always "well bounded." However, the adversarial noise incurred in $\mathsf{Error}_1$ can sometimes destroy the convergence of SGD. We therefore use $\{A_i\}_i$ and $\{B_i\}_i$ to perform a reasonable estimation of $\mathsf{Error}_1$, and remove the bad machines if they misbehave. Note that even at the end of the algorithm, $\mathsf{good}_T$ may still contain some Byzantine machines; however, their adversarial noise must be negligible and shall not impact the performance of the algorithm.

We have the following argument to establish bounds on the two error terms:

**Lemma 3.1.** *With probability $1 - \delta$, we simultaneously have*

$$\big|\mathsf{Error}_1\big| \leq 4D\mathcal{V}\sqrt{TmC} + 16\alpha mD\mathcal{V}\sqrt{TC} \quad \text{and} \quad \mathsf{Error}_2 \leq 32\alpha^2\mathcal{V}^2 + \frac{4\mathcal{V}^2 C}{m} \; .$$

**Algorithm 1** `ByzantineSGD`$(\eta, x_1, D, T, \mathfrak{T}_A, \mathfrak{T}_B)$

---

**Input:** learning rate $\eta > 0$, starting point $x_1 \in \mathbb{R}^d$, diameter $D > 0$, number of iterations $T$, thresholds $\mathfrak{T}_A, \mathfrak{T}_B > 0$;

$\quad\quad\quad \diamond$ *theory suggests $\mathfrak{T}_A = 4D\mathcal{V}\sqrt{T\log(16mT/\delta)}$ and $\mathfrak{T}_B = 4\mathcal{V}\sqrt{T\log(16mT/\delta)}$*

$\quad\quad\quad\quad\quad\quad\quad\quad \diamond$ *where $\delta$ is confidence parameter*

1: $\mathsf{good}_1 \leftarrow [m]$;
2: **for** $k \leftarrow 1$ **to** $T$ **do**
3: $\quad$ **for** $i \leftarrow 1$ **to** $m$ **do**
4: $\quad\quad$ receive $\nabla_{k,i} \in \mathbb{R}^d$ from machine $i \in [m]$; $\quad\quad \diamond$ *we have $\mathbb{E}[\nabla_{k,i}] = \nabla f(x_k)$ if $i \in \mathsf{good}$*
5: $\quad\quad$ $A_i \leftarrow \sum_{t=1}^{k}\langle \nabla_{t,i}, x_t - x_1\rangle$ and $B_i \leftarrow \sum_{t=1}^{k} \nabla_{t,i}$;
6: $\quad$ **end for**
7: $\quad$ $A_{\mathsf{med}} := \mathsf{median}\{A_1, \dots, A_m\}$
8: $\quad$ $B_{\mathsf{med}} \leftarrow B_i$ where $i \in [m]$ is any machine s.t. $\big|\{j \in [m]: \|B_j - B_i\| \leq \mathfrak{T}_B\}\big| > m/2$.

$\quad\quad\quad\quad \diamond$ *all machines $i \in \mathsf{good}$ will be valid choice, see Claim A.3b*

9: $\quad$ $\nabla_{\mathsf{med}} \leftarrow \nabla_{k,i}$ where $i \in [m]$ is any machine s.t. $\big|\{j \in [m]: \|\nabla_{k,j} - \nabla_{k,i}\| \leq 2\mathcal{V}\}\big| > m/2$

$\quad\quad\quad\quad \diamond$ *all machines $i \in \mathsf{good}$ will be valid choice due to Assumption 2.2*

10: $\quad$ $\mathsf{good}_k \leftarrow \big\{i \in \mathsf{good}_{k-1}: |A_i - A_{\mathsf{med}}| \leq \mathfrak{T}_A \wedge \|B_i - B_{\mathsf{med}}\| \leq \mathfrak{T}_B \wedge \|\nabla_{k,i} - \nabla_{\mathsf{med}}\| \leq 4\mathcal{V}\big\}$;

$\quad\quad\quad\quad \diamond$ *with high probability $\mathsf{good}_k \supseteq \mathsf{good}$*

11: $\quad$ $x_{k+1} = \arg\min_{y:\,\|y-x_1\|\leq D}\big\{\frac{1}{2}\|y - x_k\|^2 + \eta\langle\frac{1}{m}\sum_{i\in\mathsf{good}_k}\nabla_{k,i}, y - x_k\rangle\big\}$;
12: **end for**

---

The proof of this lemma will be in two parts: first, we define a set of determinstic conditions, and show that these conditions hold with high probability. Then, we will demonstrate that assuming these concentration results hold, the error will be bounded. The details of the proof are deferred to the full version of this paper.

With this crucial lemma, we can now prove some rates for our algorithm.

**Smooth functions.** We first consider the setting where our objective is smooth, and prove:

**Theorem 3.2.** *Suppose in Problem (3.1) our $f(x)$ is L-smooth and Assumption 2.2 holds. Suppose $\eta \leq \frac{1}{2L}$ and $\mathfrak{T}_A = 4D\mathcal{V}\sqrt{TC}$ and $\mathfrak{T}_B = 4\mathcal{V}\sqrt{TC}$. Then, with probability at least $1 - \delta$, letting $C := \log(16mT/\delta)$ and $\overline{x} := \frac{x_2 + \cdots + x_{T+1}}{T}$, we have*

$$f(\overline{x}) - f(x^*) \leq \frac{D^2}{\eta T} + \frac{8D\mathcal{V}\sqrt{TmC} + 32\alpha m D\mathcal{V}\sqrt{TC}}{Tm} + \eta \cdot \Big(\frac{8\mathcal{V}^2 C}{m} + 32\alpha^2\mathcal{V}^2\Big)\ .$$

*If $\eta$ is chosen optimally, then*

$$f(\overline{x}) - f(x^*) \leq O\Big(\frac{LD^2}{T} + \frac{D\mathcal{V}\sqrt{C}}{\sqrt{Tm}} + \frac{\alpha D\mathcal{V}\sqrt{C}}{\sqrt{T}}\Big)\ .$$

We remark that

- The first term $O\big(\frac{LD^2}{T}\big)$ is the classical error rate for gradient descent on smooth objectives [21] and should exist even if $\mathcal{V} = 0$ (so every $\nabla_{k,i}$ exactly equals $\nabla f(x_k)$) and $\alpha = 0$.
- The first two terms $\widetilde{O}\big(\frac{LD^2}{T} + \frac{D\mathcal{V}}{\sqrt{Tm}}\big)$ together match the classical mini-batch error rate for SGD on smooth objectives, and should exist even if $\alpha = 0$ (so we have no Byzantine machines).
- The third term $\widetilde{O}\big(\frac{\alpha D\mathcal{V}}{\sqrt{T}}\big)$ is optimal in our Byzantine setting due to Theorem 4.3.

*Proof of Theorem 3.2.* Applying Fact 2.4 for $k = 1, 2, \dots, T$ with $u = x^*$, we have

$$\frac{1}{T}\sum_{k\in[T]}\langle\xi_k, x_k - x^*\rangle \leq \frac{D^2}{2\eta T} + \frac{1}{T}\sum_{k\in[T]}\Big(\langle\xi_k, x_k - x_{k+1}\rangle - \frac{1}{2\eta}\|x_k - x_{k+1}\|^2\Big)$$

$$= \frac{D^2}{2\eta T} + \frac{1}{T}\sum_{k\in[T]}\Big(\Big\langle\frac{1}{m}\sum_{i\in\mathsf{good}_k}\nabla_{k,i}, x_k - x_{k+1}\Big\rangle - \frac{1}{2\eta}\|x_k - x_{k+1}\|^2\Big)$$

$$(3.2)$$

We notice that the left hand side of (3.2)

$$\sum_{k\in[T]} \langle \xi_k, x_k - x^* \rangle \tag{3.3}$$

$$= \frac{1}{m} \sum_{k\in[T]} \sum_{i\in\mathsf{good}_k} \langle \nabla_k, x_k - x^* \rangle + \frac{1}{m} \sum_{k\in[T]} \sum_{i\in\mathsf{good}_k} \langle \nabla_{k,i} - \nabla_k, x_k - x^* \rangle$$

$$\overset{①}{\geq} \frac{1}{m} \sum_{k\in[T]} \sum_{i\in\mathsf{good}_k} \big(f(x_k) - f(x^*)\big) + \frac{\mathsf{Error}_1}{m}$$

$$\overset{②}{\geq} \frac{1}{m} \sum_{k\in[T]} \sum_{i\in\mathsf{good}_k} \big(f(x_{k+1}) - f(x^*) - \langle \nabla_k, x_{k+1} - x_k \rangle - \frac{L}{2}\|x_k - x_{k+1}\|^2\big) + \frac{\mathsf{Error}_1}{m} \tag{3.4}$$

Above, inequality ① uses the convexity of $f(\cdot)$ and the definition of $\mathsf{Error}_1$, and inequality ② uses the smoothness of $f(\cdot)$ which implies $f(x_{k+1}) \leq f(x_k) + \langle \nabla f(x_k), x_{k+1} - x_k \rangle + \frac{L}{2}\|x_k - x_{k+1}\|^2$.
Putting (3.4) back to (3.2), we have

$$\frac{1}{Tm} \sum_{k\in[T]} \sum_{i\in\mathsf{good}_k} \big(f(x_{k+1}) - f(x^*)\big)$$

$$\leq \frac{D^2}{2\eta T} - \frac{\mathsf{Error}_1}{Tm} + \frac{1}{T} \sum_{k\in[T]} \Big(\Big\langle \frac{1}{m} \sum_{i\in\mathsf{good}_k} (\nabla_{k,i} - \nabla_k), x_k - x_{k+1} \Big\rangle - \big(\frac{1}{2\eta} - \frac{L}{2}\big)\|x_k - x_{k+1}\|^2\Big)$$

$$\overset{①}{\leq} \frac{D^2}{2\eta T} - \frac{\mathsf{Error}_1}{Tm} + \frac{\eta}{T} \sum_{k\in[T]} \Big\|\frac{1}{m} \sum_{i\in\mathsf{good}_k} (\nabla_{k,i} - \nabla_k)\Big\|^2 = \frac{D^2}{2\eta T} - \frac{\mathsf{Error}_1}{Tm} + \eta\mathsf{Error}_2 \ . \tag{3.5}$$

Above, inequality ① uses the fact that $\frac{1}{2\eta} - \frac{L}{2} \geq \frac{1}{4\eta}$, and Young's inequality which says $\langle a, b \rangle - \frac{1}{2}\|b\|^2 \leq \frac{1}{2}\|a\|^2$.
Finally, we conclude the proof by plugging Lemma 3.1 and the following convexity inequality into (3.5):

$$\frac{1}{Tm} \sum_{k\in[T]} \sum_{i\in\mathsf{good}_k} \big(f(x_{k+1}) - f(x^*)\big) = \frac{1}{T} \sum_{k\in[T]} \frac{|\mathsf{good}_k|}{m} \big(f(x_k) - f(x^*)\big)$$

$$\geq \frac{1}{T} \sum_{k\in[T]} \frac{1}{2}\big(f(x_k) - f(x^*)\big) \geq \frac{1}{2}\big(f(\overline{x}) - f(x^*)\big) \ .$$

$\square$

**Nonsmooth Functions.** We also derive a similarly tight result when the objective is not assumed to be smooth. The proof is similar to the previous one and we defer it to the supplementary material.

**Theorem 3.3.** *Suppose in Problem (3.1) our $f(x)$ is differentiable, G-Lipschitz continuous and Assumption 2.2 holds. Suppose $\eta > 0$ and $\mathfrak{T}_A = 4D\mathcal{V}\sqrt{TC}$ and $\mathfrak{T}_B = 4\mathcal{V}\sqrt{TC}$. Then, with probability at least $1 - \delta$, letting $C := \log(16mT/\delta)$ and $\overline{x} := \frac{x_1 + \cdots + x_T}{T}$, we have*

$$f(\overline{x}) - f(x^*) \leq \frac{D^2}{\eta T} + \frac{2\eta G^2}{T} + \frac{8D\mathcal{V}\sqrt{TmC} + 32\alpha mD\mathcal{V}\sqrt{TC}}{Tm} + \eta \cdot \Big(\frac{8\mathcal{V}^2 C}{m} + 32\alpha^2\mathcal{V}^2\Big) \ .$$

*If $\eta$ is chosen optimally, then*

$$f(\overline{x}) - f(x^*) \leq O\Big(\frac{GD}{\sqrt{T}} + \frac{D\mathcal{V}\sqrt{C}}{\sqrt{Tm}} + \frac{\alpha D\mathcal{V}\sqrt{C}}{\sqrt{T}}\Big) \ .$$

We remark that, as for Theorem 3.2, the first two terms are asymptotically tight for SGD in this setting, and the last term is necessary in our Byzantine setting, as we show in Theorem 4.3.

**Strongly convex functions.** We now consider the problem[3]

$$\min_{x\in\mathbb{R}^d} \big\{f(x) := \mathbb{E}_{s\sim\mathcal{D}}[f_s(x)]\big\} \quad \text{where } f(x) \text{ is } \sigma\text{-strongly convex.} \tag{3.6}$$

In this setting, we can obtain similarly optimal rates to those we obtained before, by reducing the problem to repeatedly solving non-strongly convex ones, as in Hazan and Kale [14]. When the function is additionally smooth, we obtain:

**Theorem 3.4.** *Suppose in Problem (3.6) our $f(x)$ is L-smooth and Assumption 2.2 holds. Given $x_0 \in \mathbb{R}^d$ with guarantee $\|x_0 - x^*\| \leq D$, one can repeatedly apply* `ByzantineSGD` *to find a point $x$ satisfying with probability at least $1 - \delta'$, $f(x) - f(x^*) \leq \varepsilon$ and $\|x - x^*\|^2 \leq 2\varepsilon/\sigma$ in*

$$T = \widetilde{O}\Big(\frac{L}{\sigma} + \frac{\mathcal{V}^2}{m\sigma\varepsilon} + \frac{\alpha^2\mathcal{V}^2}{\sigma\varepsilon}\Big)$$

*iterations, where the $\widetilde{O}$ notation hides logarithmic factors in $D, m, L, \mathcal{V}, \sigma^{-1}, \varepsilon^{-1}, \delta^{-1}$.*

When the function is non-smooth, we instead obtain:

**Theorem 3.5.** *Suppose in Problem (3.6) our $f(x)$ is differentiable, G-Lipschitz continuous and Assumption 2.2 holds. Given $x_0 \in \mathbb{R}^d$ with guarantee $\|x_0 - x^*\| \leq D$, one can repeatedly apply* `ByzantineSGD` *to find a point $x$ satisfying with probability at least $1 - \delta'$, $f(x) - f(x^*) \leq \varepsilon$ and $\|x - x^*\|^2 \leq 2\varepsilon/\sigma$ in*

$$T = \widetilde{O}\Big(\frac{G^2}{\sigma\varepsilon} + \frac{\mathcal{V}^2}{m\sigma\varepsilon} + \frac{\alpha^2\mathcal{V}^2}{\sigma\varepsilon} + 1\Big)$$

*iterations, where the $\widetilde{O}$ notation hides logarithmic factors in $D, m, L, \mathcal{V}, \sigma^{-1}, \varepsilon^{-1}, \delta^{-1}$.*

We defer the proofs to the supplementary material, but we remark that again in all of these equations, our rates have three terms. Just as in the rates for non-strongly convex functions, the first two terms are necessary even when there are no Byzantine workers, and the last term matches the lower bound we give in Theorem 4.4 for Byzantine optimization.

## 4 Lower Bounds for Byzantine Stochastic Optimization

In this section, we prove that the convergence rates we obtain in Section 3 are optimal up to log factors, even in $d = 1$ dimension. Recall a random vector $X \in \mathbb{R}^d$ is subgaussian with variance proxy $\mathcal{V}^2$ if $u^T X$ is a univariate subgaussian random variable with variance proxy $\mathcal{V}^2$ for all unit vectors $u \in \mathbb{R}^d$. We require the following definition:

**Definition 4.1** (Stochastic estimator). *Given $\mathcal{X} \subseteq \mathbb{R}^d$ and $f: \mathcal{X} \to \mathbb{R}$, we say a random function $f_s$ (with $s$ drawn from some distribution $\mathcal{D}$) is a* stochastic estimator *for $f$ if $\mathbb{E}[f_s(x)] = f(x)$ for all $x \in \mathcal{X}$. Furthermore, we say $f_s$ is* subgaussian with variance proxy $\mathcal{V}^2$ *if $\nabla f_s(x) - \nabla f(x)$ is a subgaussian random variable with variance proxy $\mathcal{V}^2/d$ for all $x \in \mathcal{X}$.*

Note that the normalization factor of $1/d$ in this definition ensures that $\mathbb{E}\left[\|\nabla f_s(x) - \nabla f(x)\|^2\right] \leq O(\mathcal{V}^2)$, which matches the normalization used in this paper and throughout the literature. However, in our lower bound constructions it turns out that it suffices to take $d = 1$.

We prove our lower bounds only against subgaussian stochastic estimators. This is different from our Assumption 2.2 used in the upper-bound theorems, where we assumed $\|\nabla f_s(x) - \nabla f(x)\| \leq \mathcal{V}$ is uniformly bounded for all $x$ in the domain.

*Remark* 4.2. Such difference is negligible, because by concentration, if $f_s$ is a sample from a subgaussian stochastic estimator with variance proxy $\mathcal{V}^2$, then $\|\nabla f_s(x) - \nabla f(x)\| \leq O(\mathcal{V}\sqrt{\log(mT)})$ with overwhelming probability. As a result, this impacts our lower bounds only by a $\log(mT)$ factor. For simplicity of exposition, we only state our theorems in subgaussian stochastic estimators.

Our result for non-strongly convex stochastic optimization is the following:

**Theorem 4.3.** *For any $D, \mathcal{V}, \varepsilon > 0$ and $\alpha \in (0, 0.1)$, there exists a linear function $f: [-D, D] \to \mathbb{R}$ (of Lipscthiz continuity $G = \varepsilon/D$) with a subgaussian stochastic estimator with variance proxy $\mathcal{V}^2$ so that, given $m$ machines, of which $\alpha m$ are Byzantine, and $T$ samples from the stochastic estimator per machine, no algorithm can output $x$ so that $f(x) - f(x^*) < \varepsilon$ with probability $\geq 2/3$ unless $T = \Omega\left(\frac{D^2\mathcal{V}^2}{\varepsilon^2 m} + \frac{\alpha^2\mathcal{V}^2D^2}{\varepsilon^2}\right)$, where $x^* = \arg\min_{x \in [-D, D]} f(x)$.*

Observe that up to log factors, this matches the upper bound in Theorem 3.3 exactly, demonstrating that both are exactly tight. We get a similarly tight result for the strongly convex case:

**Theorem 4.4.** *For any $\mathcal{V}, \sigma > 0$ and $\alpha \in (0, 0.1)$, there exists a $\sigma$-strongly convex quadratic function $f : \mathbb{R} \to \mathbb{R}$ with a subgaussian stochastic estimator of variance proxy $\mathcal{V}^2$ so that, given $m$ machines, of which $\alpha m$ are Byzantine, and $T$ samples from the stochastic estimator per machine, no algorithm can output $x$ so that $|x - x^*| < \widehat{\varepsilon}$ with probability $\geq 2/3$ unless $T = \Omega\left(\frac{\mathcal{V}^2}{m\sigma^2\widehat{\varepsilon}^2} + \frac{\alpha^2\mathcal{V}^2}{\sigma^2\widehat{\varepsilon}^2}\right)$, where $x^* = \arg\min_{x \in \mathbb{R}} f(x)$.*

Since $f(x) - f(x^*) \leq \varepsilon = \frac{\sigma\widehat{\varepsilon}^2}{2}$ implies $\|x - x^*\| \leq \widehat{\varepsilon}$ by the strong convexity of $f$, Theorem 4.4 also implies the following corollary for function value approximation:

**Corollary 4.5.** *In the same setting as Theorem 4.4, no algorithm can output $x$ so that $f(x) - f(x^*) \leq \varepsilon$ with probability $\geq 2/3$ unless $T = \Omega\left(\frac{\mathcal{V}^2}{m\sigma\varepsilon} + \frac{\alpha^2\mathcal{V}^2}{\sigma\varepsilon}\right)$.*

*Remark* 4.6. The lower bound of Yin et al. [31] uses essentially the same construction as we do in the proof of Theorem 4.4. However, in $d$ dimensions, they use a subgaussian estimator for $f$ with variance proxy $d\mathcal{V}^2$ (so $\mathbb{E}\left[\|\nabla f_s(x) - \nabla f(x)\|^2\right] \leq O(d\mathcal{V}^2)$). As a result, their lower bound appears to have an additional $d$ factor in it. Once re-normalized to have variance proxy $\mathcal{V}^2$, the hard instance in [31] yields exactly the same lower bound as our Theorem 4.4.

## 5   Conclusion

We have presented the first tight (up to logarithmic factors) sample and time complexity bounds for distributed SGD in the Byzantine setting, by leveraging concentration bounds to obtain a new set of detection criteria for malevolent machines. While this setting is arguably the most fundamental setting for Byzantine SGD, there remain a number of open questions to explore. For instance, our methods require strong concentration of the gradients, strong enough to invoke Pinelis' 1994 inequality. Is it possible to achieve similar results while assuming weaker assumptions on the gradients? Alternatively, is it possible that the problem provably becomes more difficult?

There are two additional interesting questions for future work. The first is to study Byzantine-resilient variants of our protocol in a *decentralized* model, where there is no "correct" central coordinator, which can safely aggregate gradients. A second important question is exploring practical implementations of our algorithm. Our algorithm only requires adding simple, efficiently implementable checks to traditional mini-batch SGD. As a result, we believe that in practice, our algorithm should add minimal overhead, while providing strong robustness guarantees against machine failure, essentially "for free". We leave such a real-world evaluation of our method to future work.

Finally, we believe that the general algorithmic framework developed in this paper may find further applications to robust distributed estimation problems. Philosophically, our algorithm enforces conditions on the malicious machines in an "online" fashion, as the data arrives in every iteration. This is in contrast to previous approaches to Byzantine optimization such as [31] which instead enforce similar conditions using "offline" techniques, i.e. by looking at the entire dataset. The main advantage of our technique is that the per iteration time complexity is substantially faster, since we do not need to inspect the entire dataset every time. It is an interesting question whether similar techniques can yield fast distributed algorithms for other estimation problems.

## Acknowledgement

We would like to thank Yuval Peres for suggesting reference [23]. Jerry Li is supported by NSF CAREER Award CCF-1453261, CCF-1565235, a Google Faculty Research Award, and an NSF Graduate Research Fellowship.

## Footnotes

[2]This is so because even in unconstrained setting, classical SGD requires knowing an upper bound $D$ to $\|x_1 - x^*\|$ in order to choose the learning rate. We can thus add the constraint to the objective.

[3]To present the simplest result, we have assumed that Problem (3.6) is unconstrained. One can also impose an addition constraint $\|x - x_0\| \leq D$ but we refrain from doing so.

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
