[Reviews · NeurIPS 2018]

Reviewer 1



[Summary] This paper studies stochastic gradient descent to solve a convex function minimization problem in which, at each time t=1,...,T, a master algorithm selects a point x_t and gets objective function's noisy gradients at x_t from m workers, \alpha m of which are Byzantine who can return any value. The authors proposed algorithm ByzantineSGD and proved i) upper bounds on the difference between the objective function values at the mean of the selected points {x_t} and the optimal point in the case with smooth or Lipschitz continuous objective functions, ii) sample upper bounds to obtain \epsilon-close objective function values in the case with strongly convex smooth or Lipschitz continuous objective functions, and iii) sample lower bounds in the case with non-strong convex or strong convex objective functions. [Originality] The same Byzantine convex function minimization problem has been independently considered by Yin et al. [31] but in their setting, each worker returns the gradient averaged over n samples, which is not SGD but GD. Their algorithm is completely different from ByzantineSGD. [Strengths] * First SGD algorithm for Byzantine setting * Nice theoretical analyses * Comprehensive theoretical sample complexity comparison with related algorithms [Weakness] * The same optimization problem has been independently studied using a similar approach. * The master algorithm cannot be seen as SGD. * No empirical comparison with the existing algorithms [Recommendation] I recommend this paper to be evaluated as "Marginally above the acceptance threshold". This is a theoretical paper and no empirical result is shown. Theoretical comparison with existing algorithms has been done comprehensively. One anxiety is practical usefulness of this algorithm. SGD needs many iteration until convergence, so communication cost between the master and workers may become larger than GD. [Detailed Comments] p.5 \Delta_i, \Delta_{med} and \nabla_i are not defined.

Reviewer 2



The paper studies stochastic convex optimization in a distributed master/workers framework, where on each round each machine out of m produces a stochastic gradient and sends it to the master, which aggregates these into a mini-batch. In this paper the authors allow a fraction of alpha of the machines to be Byzantine, i.e., they do not need to report valid stochastic gradients but may produce arbitrary vectors, even in an adversarial manner. The goal is to aggregate the reports of the machines and to converge to an optimal solution of the convex objective despite the malicious Byzantine machines. The authors present a novel variant of minibatch-SGD which tackles the difficulty the dealing with Byzantine machines. They prove upper-bounds on the convergence and nearly optimal matching lower-bounds on any algorithm working in such framework, and in this sense the results are quite satisfactory. Importantly, the convergence is roughly the same as mini-batch SGD but includes an additional natural term that accounts for the Byzantine machines. A somewhat similar result appeared in ICML18 [31]. [31] considers a very related problem, yet different, in which each machine is not represented by a stochastic first-order oracle, but is given a batch of m samples for the population distribution. The authors in [31] give a master/workers algorithm not based on SGD but on deterministic GD, where on each round, each machine reports the gradient w.r.t. its entire batch. Both the current work and [31] can be compared w.r.t to sample complexity, i.e., number of samples each machine uses to guarantee a certain approximation error. While the two works use somewhat different assumptions, still the current work improves a key term in the convergence rate in terms of the dimension of the problem (linear dependence on d in [31] vs. no dependence in current paper). I find this improvement to be of great importance, especially for large-scale problems. Moreover, the algorithmic techniques used in this current paper seems to be quite different than those in [31], e.g., current paper tries to track the set of non-Byzntine machines throughout the run, while no such argument is present in [31]. Hence, also in terms of techniques the current paper seems sufficiently novel.

Reviewer 3



The paper tackles a recently studied question in (parallel) optimization: robustness to adversarial gradients. It fits quite well in the growing body of work on the topic and presents an optimality results (although there is a significan overlap with what was proven by Yin et al in ICML 2018: https://arxiv.org/abs/1803.01498 ) Strength: optimality, information-theoretical limit, elegant (and easily verifiable) proofs. Weakness: - only valid for convex loss functions. Although I would fight for the relevance of convex optimization, in the context of poisoning attacks, the main challenges lies in what an attacker could do because the loss function is not convex. Not to mention that the state of the art models are non-convex (deep learning for instance), in this case, the solution of the authors not only cannot be used as is, but I doubt it has any valid adaptation: each quorum will keep optimizing far away from the others, each in a distinct valley of the loss function. The learned models by each quorum will therefore be impossible to aggregate in a useful manner. -relevant to the previous point: complete absence of experimental evaluation (not even in the convex case), this could have illustrated the claimed speed guarantees, which are the central claims of the paper. I enjoyed reading the proofs in the paper and found them interesting for their own sake. I would however be happier if the authors provided experimental illustration, and considered the more interesting/relevant case of non-convex optimization, or at least hinted at how one could transpose their solution there. So far I could only see the aforementioned limitation on quorum that makes it impossible to re-use this in practical non-convex ML. In that sense, the conclusion of the paper is abruptly short, more efforts should be put by the authors there than a mere "Avenues for future work include exploring practical implementations of our algorithm".